# Healthful Eating Behaviors among Couples Contribute to Lower Gestational Weight Gain

**DOI:** 10.3390/nu16060822

**Published:** 2024-03-13

**Authors:** Joshua R. Sparks, Leanne M. Redman, Kimberly L. Drews, Clark R. Sims, Rebecca A. Krukowski, Aline Andres

**Affiliations:** 1Pennington Biomedical Research Center, Louisiana State University, Baton Rouge, LA 70808, USA; joshua.r.sparks@leidos.com (J.R.S.); leanne.redman@pbrc.edu (L.M.R.); kimberly.drews@pbrc.edu (K.L.D.); 2Expeditionary and Cognitive Sciences Research Group, Department of Warfighter Performance, Naval Health Research Center, Leidos Inc. (Contract), San Diego, CA 92152, USA; 3Arkansas Children’s Nutrition Center, Department of Pediatrics, University of Arkansas for Medical Sciences, Little Rock, AR 72205, USA; crsims@uams.edu; 4School of Medicine, University of Virginia, Charlottesville, VA 22903, USA; bkrukowski@virginia.edu

**Keywords:** couples, eating behavior, gestational weight gain, pregnancy

## Abstract

Through longitudinal analysis from the GLOWING cohort study, we examined the independent and joint relationships between couples’ eating behaviors and gestational weight gain (GWG). Pregnant persons (n = 218) and their non-pregnant partners (n = 157) completed an Eating Inventory. GWG was calculated as gestation weight at 36 weeks minus that at 10 weeks. General linear models were used to examine the relationships between GWG and the pregnant persons, non-pregnant partners, and couples (n = 137; mean of pregnant persons and non-pregnant partners) cognitive restraint (range 0–21), dietary disinhibition (range 0–18), and perceived hunger (range 0–14), with higher scores reflecting poorer eating behaviors. The adjusted models included race/ethnicity, education, income, marital status, and age. The pregnant persons and their non-pregnant partners’ cognitive restraint, dietary disinhibition, and perceived hunger scores were 9.8 ± 4.7, 4.8 ± 3.2, and 4.4 ± 2.5 and 6.6 ± 4.6, 5.4 ± 3.4, and 4.7 ± 3.2, respectively. Higher cognitive restraint scores among the pregnant persons and couples were positively associated with GWG (*p* ≤ 0.04 for both). Stratified analyses revealed this was significant for the pregnant persons with overweight (*p* ≤ 0.04). The non-pregnant partners’ eating behaviors alone were not significantly associated with GWG (*p* ≥ 0.31 for all). The other explored relationships between GWG and the couples’ eating behaviors were insignificant (*p* ≥ 0.12 for all). Among the pregnant persons and couples, reduced GWG may be achieved with higher levels of restrained eating. Involving non-pregnant partners in programs to optimize GWG may be beneficial.

## 1. Introduction

Gestational weight gain (GWG) is closely monitored throughout pregnancy to lower the risks of gestational diabetes mellitus (GDM), pre-eclampsia, a cesarean section, and infant macrosomia [1,2,3,4]. According to the National Academy of Medicine (NAM) report in 2009, GWG associated with the lowest risk of adverse maternal and neonatal health outcomes is dependent on pre-gravid BMI [5,6]. Programs during pregnancy centered around diet, which include individual-level considerations for dietary intake and eating behaviors, have been shown to be the most effective interventions to reduce GWG [7,8]. As excess GWG is associated with more energy intake compared with energy expenditure [9,10,11,12], it is imperative to understand how eating behaviors contribute to GWG to advance effective program strategies to achieve the recommended GWG.

A pregnant person’s health behaviors (e.g., eating behaviors) contribute to GWG, fetal adipose tissue development, and offspring adiposity at birth [13,14,15,16]. A prior observational study found that pregnant persons alter their eating behaviors when pregnant, including snacking and eating more food throughout the day, compared to when they are not pregnant [17]. Yet, pregnant persons who demonstrate a better ability to limit their food intake when pregnant also exhibit lower levels of perceived hunger [18]. Offsprings’ health is also influenced by the non-pregnant partner’s health behaviors, including eating behaviors [19,20,21,22]. The evidence suggests that, in non-pregnant partners, the consumption of hypercaloric or high-fat diets may be causal in the etiology of obesity development in their offspring [23,24,25,26,27,28]. Yet, the influence of non-pregnant partners’ health behaviors, specifically their eating behaviors, on pregnant persons’ GWG has not been examined.

Eating behaviors can be evaluated using the Eating Inventory, a validated 51-item questionnaire that examines the following three components of eating attitudes and behaviors: cognitive restraint, dietary disinhibition, and perceived hunger [29]. Cognitive restraint measures the tendency to restrict food intake to maintain weight, dietary disinhibition examines the overeating of palatable foods, and perceived hunger evaluates the susceptibility to feelings of hunger [29]. The evidence suggests that poorer dietary disinhibition and cognitive restraint are associated with a higher body mass index (BMI) and absolute weight gain in non-pregnant women over a 20-year follow-up period [30]. In pregnancy, the relationship between eating behaviors assessed via the Eating Inventory and the odds of excess GWG were insignificant [31]. However, another examination of 248 pregnant women found that cognitive restraint was positively associated with GWG, but that the dietary disinhibition and perceived hunger scores were not [32]. As these two studies produced contradictory findings, replicating the analyses, in addition to examining the potential independent, but also additive, effect of the non-pregnant partner’s eating behaviors with GWG, are needed.

Currently, there are no studies examining the relationship between the pregnant person and their non-pregnant partner’s eating behaviors on GWG. The evidence suggests that a cohesive partnership between expectant parents can yield more beneficial outcomes for the mother and their child, including physical and mental health and well-being [33]. Shared efforts in adopting a balanced and healthy diet in pregnancy may create a synergistic positive effect and enhance the opportunity to experience optimal GWG [34]. Non-pregnant partners during pregnancy may be able to influence the household food environment and endorse positive dietary habits and eating attitudes [34,35].

Therefore, the aim of this longitudinal analysis was to examine the relationships between pregnant persons, non-pregnant partners, and couples’ eating behaviors with GWG. We tested the following overarching hypotheses: (1) the eating behaviors of the couple would be highly (e.g., greater effect) associated with reduced GWG, (2) followed by the pregnant persons’ eating behaviors alone, and (3) the eating behaviors of the non-pregnant partners would be least associated (e.g., lower effect) with reduced GWG.

## 2. Methods

### 2.1. Overview of the GLOWING Study

The purpose of the GLOWING study (NCT01131117) was to assess the effects of maternal body composition on infants’ birth weight, growth, body composition, and risk of being overweight at 2 years old [36]. Participants were recruited for this prospective longitudinal cohort study between 2011 and 2014 from central Arkansas, the United States of America. All study procedures were approved by the Institutional Review Board (IRB) at the University of Arkansas for Medical Sciences. The inclusion criteria for pregnant persons included having 18.5 ≤ BMI ≤ 35.0 kg/m^2^, being ≥21 years of age, planning a pregnancy or <10 weeks gestational age (GA), and having only 1 other child. The exclusion criteria included having conceived with the use of fertility treatments, multiple gestation, pre-existing medical conditions (e.g., diabetes mellitus, hypertension, and sexually transmitted diseases), taking medications known to influence fetal growth (e.g., glucocorticoids, insulin, and thyroid hormones), and smoking or drinking alcohol during pregnancy. The non-pregnant partners were eligible if they were willing to provide informed consent and participate in study measurements, which have been previously described [37]. The eligible participants signed an IRB-approved informed consent prior to completing any study procedures.

### 2.2. Assessment of Participant Demographics

The pregnant persons were first assessed at <10 weeks GA. The non-pregnant partners were assessed once during the pregnancy (N = 131; 27.2 ± 12.5 weeks GA) or from the time of birth to 2 years postpartum (N = 26; 31.8 ± 38.8 weeks postpartum). For this analysis, age was assessed continuously, while marital status (Married or Cohabitating, Single, Divorced, or Unknown/Not Reported), race/ethnicity (White/Caucasian; Black/African American; Asian or more than one race; Unknown/Not Reported), highest educational level completed (<College graduate; ≥College graduate; Unknown/Not Reported), and individual income (<USD 40,000 USD; ≥USD 40,000 USD; Unknown/Not Reported) were assessed categorically.

### 2.3. Pregnant Persons’ and Non-Pregnant Partners’ Weight and Body Mass Index (BMI)

For the pregnant persons, weight was measured at <10 weeks (first trimester) and 36 weeks (third trimester) GA. The weight of the non-pregnant partners was measured once. Height was measured once for each participant, and BMI (kg/m^2^) = Weight (kg) ÷ Height (m)^2^ was computed and categorized as normal weight (18.5 ≤ BMI < 25.0), overweight (25.0 ≤ BMI < 30), or obesity (30.0 ≥ BMI).

### 2.4. Gestational Weight Gain (GWG)

GWG (kg) was calculated as the difference in measured clinic weights at 36 weeks and <10 weeks GA. GWG, with adjustment for GA at the time of final weight measurement (36 weeks), was then classified as inadequate, recommended, or excess in accordance with the following BMI classifications: normal weight (recommended: 10.1–14.0 kg), overweight (recommended: 6.08–10.18 kg), or obesity (recommended: 4.32–7.92 kg) [38].

### 2.5. Pregnant Persons and Non-Pregnant Partners’ Eating Behaviors

For the pregnant persons, the Eating Inventory was administered at the first and/or third trimester of pregnancy. The Eating Inventory constructs did not change significantly throughout pregnancy (*p* = 0.86 for all, N = 80). Therefore, the pregnant persons who had Eating Inventory scores in either only the first (N = 128) or third trimester (N = 10) were included and, for those with both trimesters, the first and third trimester scores were averaged for analysis. The Eating Inventory of the partner was administered once either during their pregnancy or a postpartum visit.

Scores for cognitive restraint, dietary disinhibition, and perceived hunger ranged from 0 to 21, 0 to 18, and 0 to 14, respectively, and a greater score indicates poorer levels of each respective eating behavior [29]. Each of the three constructs were further categorized as Low (better) or High (poorer), respectively: cognitive restraint (Low ≤ 10; High > 10), dietary disinhibition (Low ≤ 8; High > 8), and perceived hunger (Low ≤ 7; High > 7) [29]. For the couple’s Eating Inventory analyses, the scores of the pregnant persons and their non-pregnant partners were averaged and categorized using the same cut-offs.

### 2.6. Statistical Analyses and Power Analysis Calculations

SAS Version 9.4 (SAS Institute; Cary, NC, USA) was used for statistical analysis. In total, 218 pregnant persons had GWG and Eating Inventory data, 157 non-pregnant partners had Eating Inventory data, and Eating Inventory data were available for 137 couples (pregnant persons and non-pregnant partners) (Appendix A).

Demographic data are presented as mean ± SD for continuous variables and as number and percentage for proportional or categorical data. A data review was performed to ensure normality of the data and to assess if there were any outliers in the data for any variables of interest (no outliers were identified in this analysis). The participant characteristics included as covariates for analysis were examined against early pregnancy BMI and assessed for collinearity to prioritize covariates in modeling.

To examine the relationship between the pregnant persons, non-pregnant partners, and couples’ eating behaviors (continuous and categorical) with GWG, we used general linear models (GLM using proc GLM statement with estimates function). Additionally, to examine the odds of excess GWG, we performed odds ratio estimation (using proc logistic statement with estimate function). The pregnant persons and non-pregnant partners’ characteristics included in the adjusted models were marital status, race/ethnicity, education, household income, and categorical pregnant person BMI at <10 weeks GA. A significance value of *p* < 0.05 was set.

Power analysis calculations were performed using G*Power 3.0.10 (Universitat Kiel, Kiel, Germany). A sample of 115 pregnant persons provides a moderate-to-high correlation (0.3 < r < 0.6) between the eating behavior constructs and GWG. With 27–62 participants ranging from having a normal weight to obesity, a large effect (r > 0.6) could be detected based on the Pearson correlation coefficient. To detect a significant relationship between the eating behavior constructs and GWG, the multiple linear regressions were adequately powered following adjustment for the planned a priori covariates (critical r^2^ = 0.16). When early pregnancy BMI was examined along with prior a priori covariates included in the model, they remained adequately powered for our planned analyses, with 49 participants required for α = 0.05 and 1 − β error probability set to 0.95.

## 3. Results

### 3.1. Overall Participant Demographics

The demographic characteristics of the pregnant persons (n = 218) and their non-pregnant partners (n = 157) are presented in Table 1. On average, the pregnant persons and their non-pregnant partners were 30.3 ± 3.7 and 31.4 ± 4.1 years of age, respectively, and had mean BMIs of 26.1 ± 4.3 and 28.5 ± 5.2 kg/m^2^, respectively. Although there were no significant differences between the pregnant persons and their non-pregnant partners’ age and race/ethnicity, a greater proportion of the non-pregnant partners were classified as a ≥College graduate, earning an individual income ≥USD 40,000 USD, and having obesity (BMI ≥ 30.0 kg/m^2^) compared to the pregnant persons (*p* ≤ 0.01 for all).

### 3.2. Gestational Weight Gain

The mean GWG of the overall cohort was 11.8 ± 4.3 kg, and the majority (48.6%) experienced excess GWG (Table 2). Most pregnant persons having overweight (63.0%) or obesity (52.2%) experienced excess GWG compared to only one-third of those having a normal weight (36.1%, *p* < 0.001).

### 3.3. Parental Eating Inventory Eating Behaviors

When examining the Eating Inventory constructs (Table 2), the pregnant persons’ average scores for cognitive restraint, dietary disinhibition, and perceived hunger were 9.8 ± 4.7 (56.9% Low), 4.8 ± 3.2 (85.8% Low), and 4.4 ± 2.5 (88.5% Low), respectively. The cognitive restraint, dietary disinhibition, and perceived hunger scores were significantly lower in the pregnant persons with a normal weight compared to those having overweight or obesity (*p* < 0.01 for all). No differences were observed between those having overweight or obesity (*p* ≥ 0.76 for all). A greater proportion of pregnant persons having overweight or obesity, compared to the pregnant persons with a normal weight, expressed categorically High (poorer) classifications of cognitive restraint, dietary disinhibition, and perceived hunger (*p* < 0.01 for all).

In the non-pregnant partners, the cognitive restraint, dietary disinhibition, and perceived hunger average scores were 6.6 ± 4.6 (80.9% Low), 5.4 ± 3.4 (85.4% Low), and 4.7 ± 3.2 (79.6% Low), respectively. There were no significant differences in eating behaviors for the non-pregnant partners who completed their study visit during pregnancy compared to those postpartum (*p* ≥ 0.53 for all; Appendix A). Of note, the cognitive restraint score was significantly higher (poorer) among the pregnant persons compared to that of their non-pregnant partner (*p* = 0.03). Similarly, a lower proportion of pregnant persons were classified as having a Low (better) cognitive restraint level compared to their non-pregnant partner (56.9% compared to 80.9%; *p* = 0.03).

### 3.4. Relationships between the Pregnant Persons’ Eating Behaviors with GWG

The relationship between the pregnant persons’ cognitive restraint, dietary disinhibition, and perceived hunger with GWG in the overall sample and according to the early pregnancy BMI are presented in Figure, Panel 1. In the unadjusted models, the continuous and categorical cognitive restraint values were positively associated with GWG (Figure 1A) (*p* = 0.02 and 0.03, respectively). The relationship between continuous cognitive restraint and GWG persisted following adjustment for early pregnancy BMI only (β = 0.2, SE = 0.1, *p* < 0.0001) and with the addition of marital status, race/ethnicity, education, and household income to the model (β = 0.2, SE = 0.1, *p* = 0.0004). When stratified by early pregnancy BMI, the relationship between continuous cognitive restraint and GWG was only observed among the women having overweight (unadjusted: β = 0.17, SE = 0.08, *p* = 0.036; adjusted: β = 0.21, SE = 0.09, *p* = 0.027; Figure 1B). No significant relationships were observed between dietary disinhibition or perceived hunger (continuous or categorical) with GWG (*p* ≥ 0.12 for all, Figure 1C–F). Interestingly, there were no significant odds of experiencing excess GWG when comparing the Low- and High-level categorizations of any eating behavior construct (*p* ≥ 0.06 for all).

### 3.5. Relationships between Non-Pregnant Partners’ Eating Behaviors with GWG

The relationship between the non-pregnant partners’ cognitive restraint, dietary disinhibition, and perceived hunger with GWG in the overall sample and BMI are presented in Figure 2. No significant relationships were observed between the non-pregnant partners’ continuous or categorical cognitive restraint, dietary disinhibition, or perceived hunger scores with the pregnant persons’ GWG (Figure 2A, Figure 2C, and Figure 2E, respectively) in the unadjusted model (cognitive restraint continuous β = 0.04, SE = 0.05 and categorical β = 0.001, SE = 0.04; dietary disinhibition continuous β = −0.04, SE = 0.11 and categorical β = −0.002, SE = 0.006; perceived hunger continuous β = −0.02, SE = 0.12 and categorical β = −0.002, SE = 0.006; *p* ≥ 0.26 for all). These findings persisted in the adjusted models (cognitive restraint continuous β = 0.06, SE = 0.08 and categorical β = 0.001, SE = 0.004; dietary disinhibition continuous β = 0.04, SE = 0.11 and categorical β = −0.002, SE = 0.006; perceived hunger continuous β = 0.07, SE = 0.12 and categorical β = 0.001, SE = 0.006; *p* ≥ 0.27 for all). There were similar associations between eating behaviors with GWG when examined according to early pregnancy BMI (Figure 2B, Figure 2D, and Figure 2F, respectively; unadjusted *p* ≥ 0.07 for all; adjusted *p* ≥ 0.06 for all). There were no significant odds of experiencing excess GWG when comparing the Low- and High-level categorizations of any eating behavior construct (*p* ≥ 0.17 for all).

### 3.6. Relationship between the Average Eating Behaviors of the Pregnant Persons and Non-Pregnant Partners with GWG

Of the overall sample, 137 pregnant persons and 137 non-pregnant partners had eating behavior data that were used to compute the mean couple scores for each eating behavior construct (the participant characteristics and eating behavior data are presented in Appendix A). The couples’ average cognitive restraint score was 8.3 ± 3.4, with 70.3% considered to have Low scores. The average dietary disinhibition score was 4.9 ± 2.3, with most of the couples (94.9%) having categorically Low scores, which aligned with the perceived hunger scores (4.6 ± 2.0; 91.2% Low).

As shown in Figure 3A, couple cognitive restraint scores were positively associated with GWG, such that for each one unit increase in cognitive restraint, there was a 0.23 (0.11) kg increase in GWG (*p* = 0.03). Furthermore, for the couples with Low versus High cognitive restraint scores, High (poorer) cognitive restraint equated to a 1.6 (0.4) kg increase in GWG (*p* = 0.04), but not an increase in the odds of experiencing excess GWG (OR: 0.87; 95% CI: 0.25, 2.12; *p* = 0.22). These relationships persisted following the adjustment for BMI (*p* ≤ 0.02 for both) and with the addition of marital status, race/ethnicity, education, and household income to the model (*p* ≤ 0.02 for both).

In the unadjusted models, no significant relationships were observed between couple dietary disinhibition or perceived hunger with GWG (*p* ≥ 0.18 for all, Figure 3C,E). There was no effect of perceived hunger on GWG when stratified by early pregnancy BMI (Figure 3F). In the adjusted models (BMI, marital status, race/ethnicity, education, and household income), the couples’ categorical dietary disinhibition scores were positively associated with GWG (Figure 3D), such that a categorically High (poorer) dietary disinhibition score for the couples equated to a 4.0 (1.4) kg (*p* = 0.005) increase in GWG. There was no significant relationship in the adjusted models between the couples’ continuous dietary disinhibition and continuous and categorical perceived hunger and GWG.

## 4. Discussion

To our knowledge, this is the first study to assess the independent and synergistic effect of pregnant persons and their non-pregnant partners’ eating behaviors on GWG. Our findings suggest that poorer cognitive restraint by pregnant persons is positively associated with increased GWG, while no significant relationships were observed in the pregnant persons or their non-pregnant partners’ dietary disinhibition or perceived hunger with GWG. Yet, poorer cognitive restraint and dietary disinhibition scores in the couple were associated with increased GWG, suggesting that poorer control over their eating behaviors is associated with increased GWG. Collectively, this evidence supports the importance of involving the pregnant couple together in behavioral intervention studies, particularly centered upon diet habits and eating behaviors, to optimize our understanding of GWG.

The pregnant persons’ dietary habits, including adequate dietary intake (e.g., sufficient kilocalorie and macronutrient intake) and dietary quality (e.g., high fruit and vegetable intake) during pregnancy have been shown to positively impact GWG [11,39]. A 2023 systematic review and meta-analysis from fourteen studies across eleven countries (n = 77,550) found that unhealthy dietary patterns, consisting of greater saturated fats and simple carbohydrates, were associated with an increased odds of experiencing excess GWG (OR = 1.22, 95% CI: 1.02–1.45, *p* = 0.031), regardless of their pre-/early pregnancy BMI [40]. Therefore, improving diet habits and eating behaviors may act as a strategy for enhancing opportunities to achieve the recommended GWG.

Eating behaviors, including cognitive restraint, dietary disinhibition, and perceived hunger assessed using the Eating Inventory, have been shown to be important for understanding the contributors to body weight status (i.e., BMI) and weight management outside of pregnancy. For example, in over 2500 adults, dietary disinhibition (the tendency to overeat palatable foods) was positively associated with BMI, with the highest cognitive restraint scores (less control over food intake) found in non-pregnant women with obesity [41]. In pregnancy, a prior examination of 248 pregnant women found that cognitive restraint was positively associated with GWG, but that the dietary disinhibition and perceived hunger scores were not [32]. However, the relationship between eating behaviors assessed using the Eating Inventory and the odds of excess GWG were previously shown to be insignificant [31]. These results align with our findings that the pregnant persons’ eating behaviors, specifically cognitive restraint, may be most impactful for weight management in pregnancy, but they do not predict the odds of experiencing excess GWG.

Whether the non-pregnant partner is an active or passive bystander in health behavior decision making during pregnancy has been debated [42]. To our knowledge there has been no evaluation comparing the dietary changes in pregnant persons to those in their non-pregnant partners. Based on the prior body of literature available, the non-pregnant partners’ health behaviors could either be exclusive or impactful to weight change during pregnancy [28]. Our findings suggest that the non-pregnant partners’ eating behaviors alone were not associated with GWG. Yet, when the eating behaviors were considered jointly for the couples, the relationship between couple cognitive restraint and GWG was stronger than for the pregnant person alone, and the relationship with dietary disinhibition became significant. As such, engaging the couple in strategies to improve or develop healthful eating behaviors may be more advantageous than addressing such behaviors by the pregnant person alone. These results expand our understanding on contributors to GWG that leads to the consideration of inclusion of non-pregnant partners in future prenatal trial designs.

To date, there has been one prenatal program involving couples with the goal of lowering the risk of GDM [43]. In comparison to the couples who received usual care, those who received the GDM education intervention experienced ~2 kg less GWG [43]. Other couples have also been included in prenatal programs aimed at smoking cessation [44], reducing adverse birth outcomes (e.g., cesarean section) [45], and postpartum depression [46]. Although these individual programs may not have tailored their approach to target individual compared to couple-based effects, this body of work underscores the unique opportunity to include the non-pregnant partners during pregnancy.

### Strengths and Limitations

The primary strength of this work is the simultaneous data collected from pregnant persons and their non-pregnant partners to allow for the independent and joint evaluations of eating behaviors with GWG. Additionally, this is the largest sample size to date to allow for a priori power analysis to perform specific analysis on the outcomes of interest, while also allowing for stratified analyses using early pregnancy BMI. The Eating Inventory is an established and validated questionnaire used to assess eating behaviors, which is a major strength in this study design. In line with the validity of the Eating Inventory instrument as a strength, the eating behavior scores in our sample population aligned with the population estimates for cognitive restraint, dietary disinhibition, and perceived hunger [47,48,49]. Lastly, this is a prospective data set, as the pregnant participants were followed over time, and data about them were collected as their pregnancies progressed.

Future analyses should consider dietary intake information to assess if less-healthy eating behaviors are tracked with a higher energy intake (e.g., kilocalories per day) in addition to higher and/or excess GWG. The prior evidence has suggested that eating behaviors predict energy intake in pregnancy [50], and, therefore, we may speculate this may be an underlying root cause for excess energy intake and GWG. Additionally, a small proportion of the Eating Inventory data on the non-pregnant partners was collected in the postpartum period, rather than during pregnancy. Yet, the eating behaviors did not significantly differ from those obtained during the gestational period or compared to the postpartum period (Appendix A). Further, a small proportion of participants exhibited categorically High (poorer) eating behavior scores (3.5–31.8% dependent on construct and analysis), limiting the ability to determine the odds of excess GWG in our sample population. However, the average eating behavior scores aligned with the prior published population estimates [47,48,49]. Lastly, other health behaviors (e.g., physical activity and sleep) contribute to GWG, which remain unaccounted for in the present analysis.

## 5. Conclusions

Our findings suggest that pregnant persons’ cognitive restraint was positively associated with GWG, such that poorer cognitive restraint was related to increased GWG. When examining the couples’ eating behaviors, the relationship between cognitive restraint and GWG was stronger than that examined for the pregnant persons only. Further, a positive relationship between couple dietary disinhibition and GWG was also significant after adjusting for the covariates, such that poorer dietary disinhibition was related to increased GWG. No significant relationships were observed between the individuals’ or couples’ perceived hunger and GWG. Collectively, these results propose that the inclusion of strategies for managing the couples’ eating behaviors (e.g., stimulus control by removing or not introducing highly palatable foods to the household) in prenatal programs may provide additive benefit for meeting the GWG recommendation, which, in turn, would improve the maternal and child outcomes. Finally, this work underscores the need to explore the role of non-pregnant partners and couples’ health behaviors during pregnancy.

## Figures and Tables

**Figure 1 nutrients-16-00822-f001:**
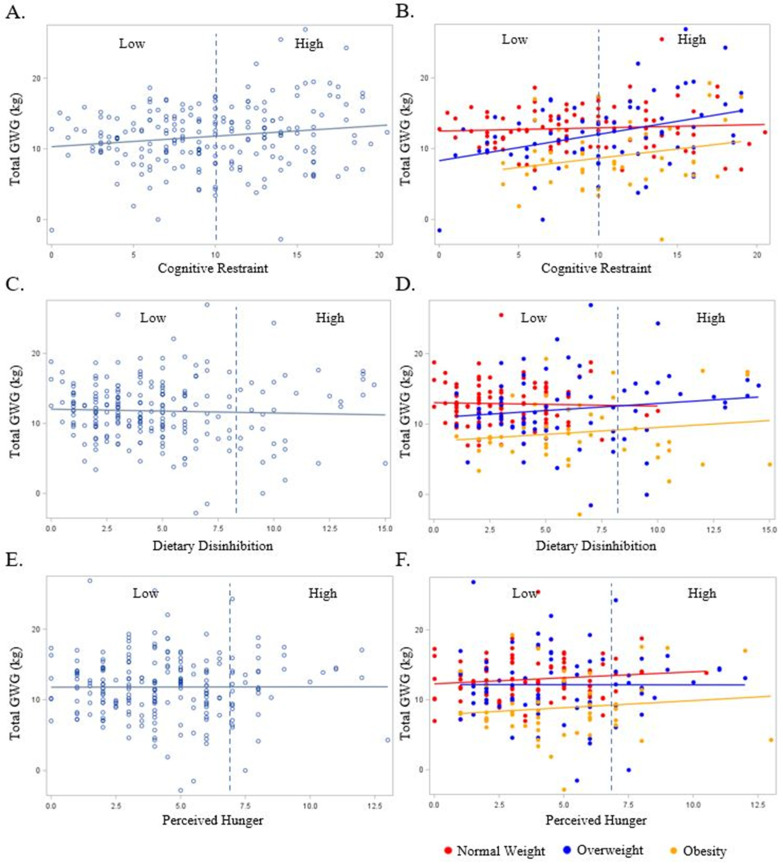
The relationship between the pregnant persons’ Eating Inventory constructs with gestational weight gain overall (light blue plot points and regression line) and BMI (red, dark blue, and yellow plot points and regression lines). BMI is designated as a normal weight (Red), overweight (Dark Blue), and obese (Yellow). Dashed line represents cut-off point between Low and High scores for each respective eating behavior construct as follows: cognitive restraint (Low ≤ 10; High > 10), dietary disinhibition (Low ≤ 8; High > 8), and perceived hunger (Low ≤ 7; High > 7). (**A**) Total gestational weight gain and cognitive restraint overall; (**B**) total gestational weight gain and cognitive restraint by BMI; (**C**) total gestational weight gain and dietary disinhibition overall; (**D**) total gestational weight gain and dietary disinhibition by BMI; (**E**) total gestational weight gain and perceived hunger overall; (**F**) total gestational weight gain and perceived hunger by BMI.

**Figure 2 nutrients-16-00822-f002:**
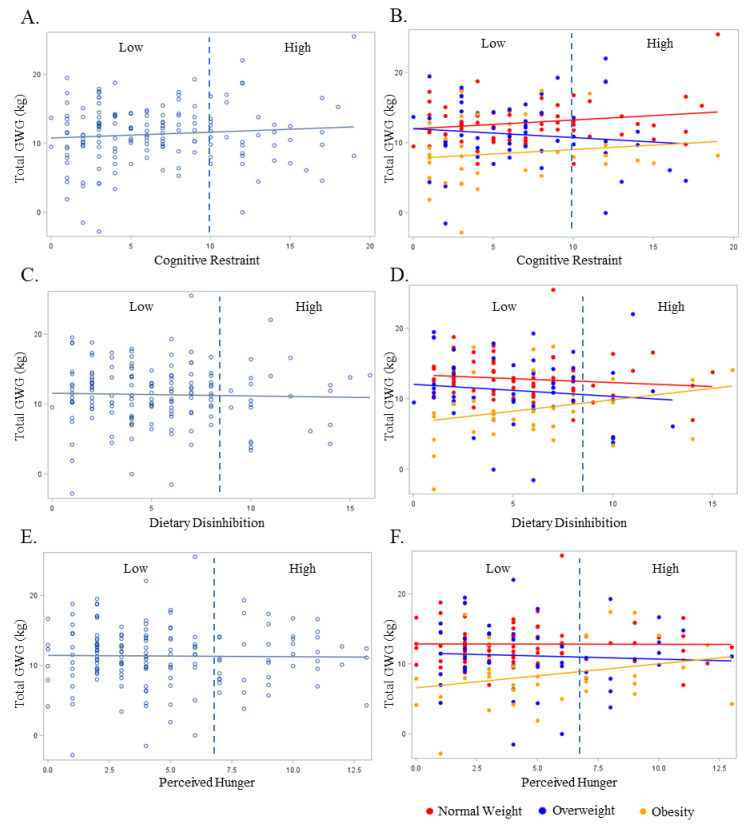
The relationship between the non-pregnancy partners’ Eating Inventory constructs with gestational weight gain overall (light blue plot points and regression line) and BMI (red, dark blue, and yellow plot points and regression lines). BMI is designated as normal weight (Red), overweight (Dark Blue), or obese (Yellow). Dashed line represents cut-off point between Low and High scores for each respective eating behavior construct as follows: cognitive restraint (Low ≤ 10; High > 10), dietary disinhibition (Low ≤ 8; High > 8), and perceived hunger (Low ≤ 7; High > 7). (**A**) Total gestational weight gain and cognitive restraint overall; (**B**) total gestational weight gain and cognitive restraint by BMI; (**C**) total gestational weight gain and dietary disinhibition overall; (**D**) total gestational weight gain and dietary disinhibition by BMI; (**E**) total gestational weight gain and perceived hunger overall; (**F**) total gestational weight gain and perceived hunger by BMI.

**Figure 3 nutrients-16-00822-f003:**
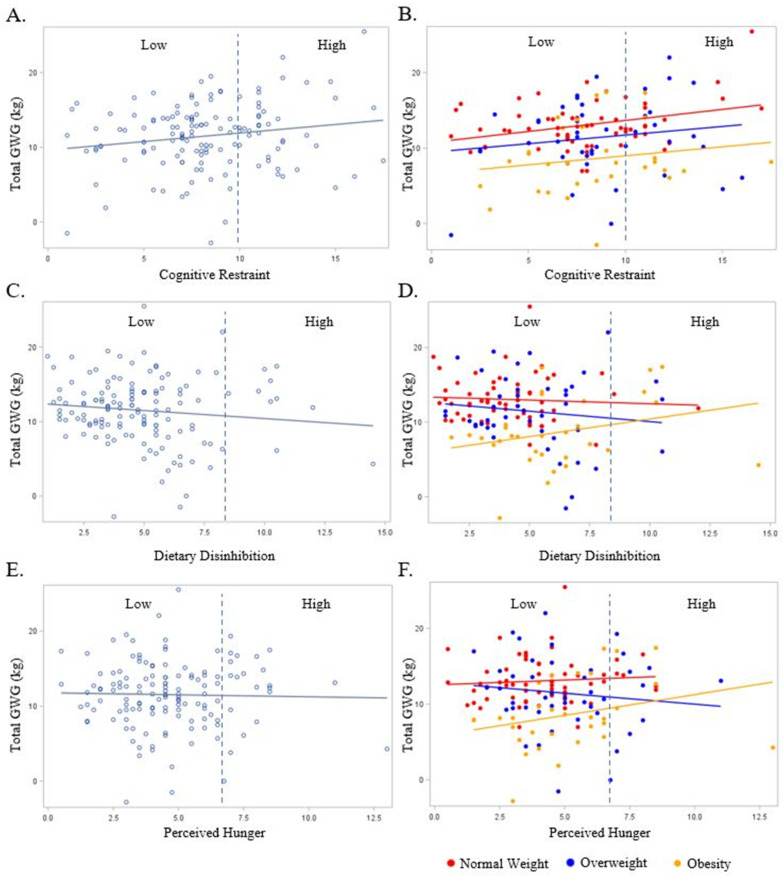
The relationship between the couples’ Eating Inventory constructs with gestational weight gain overall (light blue plot points and regression line) and by BMI (red, dark blue, and yellow plot points and regression lines). BMI is designated as normal weight (Red), overweight (Dark Blue), or obese (Yellow). Dashed line represents cut-off point between Low and High scores for each respective eating behavior construct: cognitive restraint (Low ≤ 10; High > 10), dietary disinhibition (Low ≤ 8; High > 8), and perceived hunger (Low ≤ 7; High > 7). (**A**) Total gestational weight gain and cognitive restraint overall; (**B**) total gestational weight gain and cognitive restraint by BMI; (**C**) total gestational weight gain and dietary disinhibition overall; (**D**) total gestational weight gain and dietary disinhibition by BMI; (**E**) total gestational weight gain and perceived hunger overall; (**F**) total gestational weight gain and perceived hunger by BMI.

**Table 1 nutrients-16-00822-t001:** Baseline characteristics and demographics of pregnant persons and non-pregnant partners included in analysis.

	Pregnant Person (n = 218)	Non-Pregnant Partner (n = 157)	*p*-Value
Age (Mean ± SD)	30.3 ± 3.7	31.4 ± 4.1	0.07
Body Mass Index (BMI; kg/m^2^; Mean ± SD)	26.1 ± 4.3	28.5 ± 5.2	<0.0001
Normal Weight, N (%)	99 (45.4)	44 (28.0)	<0.0001
Overweight, N (%)	73 (33.5)	62 (39.5)	
Obesity, N (%)	46 (21.1)	51 (35.5)	
Marital Status *			
Married or Cohabiting, N (%)	191 (87.6)	-	
Single, Divorced, or Unknown/Not Reported, N (%)	27 (12.4)	-	
Race/Ethnicity			0.68
White/Caucasian, N (%)	169 (77.5)	129 (82.2)	
Black/African American, N (%)	22 (10.1)	16 (10.2)	
Asian or more than One Race, N (%)	3 (1.4)	0 (0.0)	
Unknown or Not Reported, N (%)	24 (11.0)	12 (7.6)	
Highest Education Completed			<0.001
<College Graduate, N (%)	120 (55.0)	65 (41.4)	
≥College Graduate, N (%)	76 (34.9)	73 (46.5)	
Unknown/Not Reported, N (%)	22 (10.1)	19 (12.1)	
Individual Income			0.01
<USD 40,000 USD, N (%)	92 (42.2)	50 (31.9)	
≥USD 40,000 USD, N (%)	70 (32.1)	92 (58.6)	
Unknown/Not Reported, N (%)	56 (25.7)	15 (9.5)	

Reported as N (%) if not otherwise noted. * Marital status was only reported by pregnant persons.

**Table 2 nutrients-16-00822-t002:** Gestational weight gain and the eating inventory of pregnant persons and the eating inventory of non-pregnant partners’ overall and by body mass index.

Pregnant Person	Overall (218)	Normal Weight (99)	Overweight (73)	Obesity (46)
Gestational Weight Gain (Total; kg)	11.8 ± 4.3	12.9 ± 3.0 *	12.1 ± 4.9 *	8.8 ± 4.5
Inadequate, N (%)	26 (12.0)	14 (14.4) ^‡^	6 (8.2)	6 (13.0)
Recommended, N (%)	86 (39.4)	49 (49.5) ^‡^	21 (28.8)	16 (34.8)
Excess, N (%)	106 (48.6)	36 (36.1) ^‡^	46 (63.0)	24 (52.2)
The Eating Inventory				
Cognitive Restraint, M ± SD	9.8 ± 4.7 ^††^	9.0 ± 4.8 ^‡^	10.3 ± 4.8	10.7 ± 4.1
Low, N (%)	124 (56.9) ^††^	62 (62.6) ^‡^	38 (52.0)	24 (52.2)
High, N (%)	94 (43.1) ^††^	37 (37.4) ^‡^	35 (48.0)	22 (47.8)
Dietary Disinhibition, M ± SD	4.8 ± 3.2	3.2 ± 2.0 ^‡^	6.0 ± 3.4	6.5 ± 3.5
Low, N (%)	187 (85.8)	97 (98.0) ^‡^	56 (76.7)	34 (73.9)
High, N (%)	31 (14.2)	2 (2.0) ^‡^	17 (23.3)	12 (26.1)
Perceived Hunger, M ± SD	4.4 ± 2.5	3.7 ± 2.1 ^‡^	5.0 ± 2.6	4.9 ± 2.6
Low, N (%)	193 (88.5)	95 (96.0) ^‡^	58 (79.5	40 (87.0)
High, N (%)	25 (11.5)	4 (4.0) ^‡^	15 (20.5)	6 (13.0)
Non-Pregnant Partner	Overall (157)	Normal Weight (72)	Overweight (51)	Obese (34)
The Eating Inventory				
Cognitive Restraint, M ± SD	6.6 ± 4.6 ^††^	6.8 ± 4.7	6.8 ± 4.2	5.8 ± 4.8
Low, N (%)	127 (80.9) ^††^	59 (81.9)	41 (80.4)	27 (79.4)
High, N (%)	30 (19.1) ^††^	13 (18.1)	10 (19.6)	7 (20.6)
Dietary Disinhibition, M ± SD	5.4 ± 3.4	5.3 ± 3.2	5.3 ± 3.3	5.8 ± 3.8
Low, N (%)	134 (85.4)	63 (87.5)	43 (84.3)	28 (82.4)
High, N (%)	23 (14.6)	9 (12.5)	8 (15.7)	6 (17.6)
Perceived Hunger, M ± SD	4.7 ± 3.2	4.3 ± 3.1	4.8 ± 3.1	5.6 ± 3.6
Low, N (%)	125 (79.6)	61 (84.7)	40 (78.4)	24 (70.6)
High, N (%)	32 (20.4)	11 (15.3)	11 (21.6)	10 (29.4)

* *p* < 0.05 for significance compared to those having obesity. ^‡^ *p* < 0.05 for significance compared to those having overweight or obesity. ^††^ *p* < 0.05 for significance compared between maternal and paternal Eating Inventory constructs.

## Data Availability

Data may be made available upon reasonable request to the corresponding author.

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
