# Peer review of "Healthful Eating Behaviors among Couples Contribute to Lower Gestational Weight Gain"

_nutrients, 2024, doi:10.3390/nu16060822_

Round 1
Reviewer 1 Report
Comments and Suggestions for Authors
In this study authors investigate the relationships between pregnant person, non-pregnant partner, and couple eating behaviors with gestational weight gain.
The work is really interesting but it needs a minor revision;
Abstract: specify the acronym “GWG”.
Delete the word “race” throughout the manuscript and replace it with ethnicity.
Author Response
The authors appreciate the time and effort placed in your review of our manuscript.
In this study authors investigate the relationships between pregnant person, non-pregnant partner, and couple eating behaviors with gestational weight gain.
The work is really interesting, but it needs a minor revision;
Abstract: specify the acronym “GWG”.
Author response to reviewer: Defined GWG as gestational weight gain (GWG) following its first use in the abstract.
Delete the word “race” throughout the manuscript and replace it with ethnicity.
Author response to reviewer: We acknowledge and appreciate the reviewer’s suggestion. Rather than delete race throughout the manuscript and replace it with ethnicity, we included the term race/ethnicity throughout the manuscript.
Reviewer 2 Report
Comments and Suggestions for Authors
Dear Authors
I am pleased to serve as a reviewer for the original article titled "Healthful Eating Behaviors among Couples Contribute to Lower Gestational Weight Gain." This study aims to investigate the impact of non-pregnant partners' eating behavior on gestational weight gain using the Eating Inventory questionnaire. The authors include sufficient contextual information in the introduction to support the objective of the study: examining the impact of pre-pregnancy BMI on gestational weight increase, along with the effect of cognitive constraint and eating disinhibition on weight gain. In addition, they have identified a gap in this area of study, specifically the impact of the non-pregnant partner's eating habits on the pregnant individual's gestational weight gain, which has not been investigated.
Minor modifications
Abstract - In line 15, provide a detailed explanation of GWG. - In lines 26-27, provide a more explicit explanation of the word "other relations".
Methods
Present a flowchart illustrating the study design, including the overall number of participants in the cohort and the number of people specifically included in this study.
Discussion
There is a typographical error on lines 291-292.
Author Response
The authors appreciate the critical appraisal of our submitted manuscript, as well as reviewer comments to improve our manuscript.
Dear Authors
I am pleased to serve as a reviewer for the original article titled "Healthful Eating Behaviors among Couples Contribute to Lower Gestational Weight Gain." This study aims to investigate the impact of non-pregnant partners' eating behavior on gestational weight gain using the Eating Inventory questionnaire. The authors include sufficient contextual information in the introduction to support the objective of the study: examining the impact of pre-pregnancy BMI on gestational weight increase, along with the effect of cognitive constraint and eating disinhibition on weight gain. In addition, they have identified a gap in this area of study, specifically the impact of the non-pregnant partner's eating habits on the pregnant individual's gestational weight gain, which has not been investigated.
Minor modifications
Abstract - In line 15, provide a detailed explanation of GWG. - In lines 26-27, provide a more explicit explanation of the word "other relations".
Author response to reviewer: Defined GWG as gestational weight gain (GWG) following its first use in the abstract. Revised phrasing of “other relationships” to include more explicit explanation.
Methods
Present a flowchart illustrating the study design, including the overall number of participants in the cohort and the number of people specifically included in this study.
Author response to reviewer: Crafted a flowchart illustrating the overall and analytic sample, and included as Figure S1.
Discussion
There is a typographical error on lines 291-292.
Author response to reviewer: Corrected the typographical error.
Reviewer 3 Report
Comments and Suggestions for Authors
Dear Authors,
it was with great pleasure and interest that I read the article.
In principle, I have no comments on the substantive content of the article, only individual suggestions.
Please write in what country the research was conducted (not just the city).
In my opinion, the description to the table could be more detailed, on the other hand, the results presented in the table are clear to the reader.
Author Response
The authors thank the reviewer for their time and effort in reviewing our manuscript.
Dear Authors,
It was with great pleasure and interest that I read the article.
In principle, I have no comments on the substantive content of the article, only individual suggestions.
Please write in what country the research was conducted (not just the city).
Author response to reviewer: Added the United States of America as the country in which the study was conducted in addition to the general geographic location.
In my opinion, the description to the table could be more detailed, on the other hand, the results presented in the table are clear to the reader.
Author response to reviewer: Included some additional descriptive information in the title of the tables for reader clarity.